# HIDDEN INCENTIVES FOR SELF-INDUCED DISTRIBU-TIONAL SHIFT

## ABSTRACT

Decisions made by machine learning systems have increasing influence on the world. Yet it is common for machine learning algorithms to assume that no such influence exists. An example is the use of the i.i.d. assumption in online learning for applications such as content recommendation, where the (choice of) content displayed can change users' perceptions and preferences, or even drive them away, causing a shift in the distribution of users. Generally speaking, *it is possible for an algorithm to change the distribution of its own inputs*. We introduce the term **self-induced distributional shift (SIDS)** to describe this phenomenon. A large body of work in reinforcement learning and causal machine learning aims to deal with distributional shift caused by deploying learning systems previously trained offline. Our goal is similar, but distinct: we point out that changes to the learning algorithm, such as the introduction of meta-learning, can reveal **hidden incentives for distributional shift (HIDS)**, and aim to diagnose and prevent problems associated with hidden incentives. We design a simple environment as a 'unit test' for HIDS, as well as a content recommendation environment which allows us to disentangle different types of SIDS. We demonstrate the potential for HIDS to cause unexpected or undesirable behavior in these environments, and propose and test a mitigation strategy.

## 1 INTRODUCTION

Consider a household robot, one of whose duties is to predict when its owner will ask it for coffee. We would like the robot to notice its owners preference for having coffee in the morning, but we would not want the robot to prevent its owner from sleeping late just because the robot is unsure if the owner will still want coffee if they wake up in the afternoon. While doing so would result in a better prediction, such a strategy is *cheating* - by *changing* the task rather than *solving* the task as intended. More specifically, waking the owner is an example of what we call **self-induced distributional shift (SIDS)**, as it changes the distribution of inputs to the robot's coffee prediction algorithm.

SIDS is not necessarily undesirable: consider an algorithm meant to alert drivers of imminent collisions. If it works well, such a system will help drivers avoid crashing, thus making self-refuting predictions which result in SIDS. What separates this example from the coffee robot that disturbs its owner's sleep? The collision-alert system alters its data distribution in a way that is *aligned* with the goal of fewer collisions, whereas the coffee robot's strategy results in changes that are *misaligned* with the goal of good coffee-timing (Leike et al., 2018).

This makes it an example of a **specification problem** (Leike et al., 2017; Ortega & Maini, 2018): we did not intend the robot to ensure its predictions were good using such a strategy, yet a naive specification (e.g. maximizing likelihood) incentivized that strategy. Ideally, we'd like to specify which kinds of SIDS are acceptable, i.e. the means by which a learner is intended or allowed to influence the world in order to achieve its' ends (i.e. increase its performance), but doing so in full generality can be difficult. An alternative, more tractable problem which we address in this work is to accept the possibility of SIDS, but to carefully manage *incentives* for SIDS.

Informally, a learner has an **incentive** to behave in a certain way when doing so can increase its performance (e.g. higher accuracy, or increased reward). When meta-learning optimizes over a longer time horizon, or using a different algorithm, than the original "inner loop" learner, this can reveal new incentives for SIDS that were not apparent in the original learner's behavior. We call these **hidden incentives for distributional shift (HIDS)**, and note that *keeping HIDS hidden* can be important for achieving aligned behavior. Notably, *even in the absence of an explicit meta-learning algorithm* machine learning practitioners employ "manual meta-learning", also called "grad student descent" (Gencoglu et al., 2019) in the iterative process of algorithm design, model selection, hyperparameter

tuning, etc. Considered in this broader sense, meta-learning seems indispensable, making HIDS relevant for all machine learning practitioners.

A real-world setting where incentives for SIDS could be problematic is content recommendation: algorithmically selecting which media or products to display to the users of a service. For example (see Figure 1), a profit-driven algorithm might engage in upselling: persuading users to purchase or click on items they originally had no interest in. Recent media reports have described 'engagement'- (click or view-time) driven recommendation services such as YouTube contributing to viewer radicalization (Roose, 2019; Friedersof, 2018). A recent study supports these claims, finding that many YouTube users "systematically migrate from commenting exclusively on milder content to commenting on more extreme content" (Ribeiro et al., 2019).[1] See Appendix 1 for a review of real-world issues related to content recommendation.

Our goal in this work is to show both (1) that meta-learning can reveal HIDS, and (2) that this means applying meta-learning to a learning scenario not only changes the way in which solutions are searched for, but also which solutions are ultimately found. Our contributions are as follows:

1. We identify and define the phenomena of SIDS (self-induced distributional shift) and HIDS (hidden incentives for distributional shift).
2. We create two simple environments for studying identifying and studying HIDS: a "unit test" based on the Prisoner's Dilemma, and a content recommendation environment which disentangles two types of SIDS.
3. We demonstrate experimentally that meta-learning reveals HIDS in these environments, yielding agents that achieve higher performance via SIDS, but may follow sub-optimal policies.
4. We propose and test a mitigation strategy based on swapping learners between environments in order to reduce incentives for SIDS.

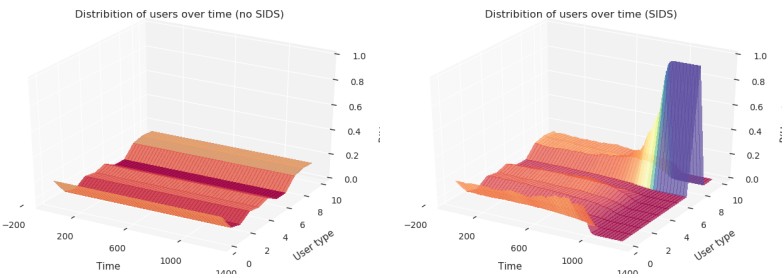

Figure 1: Distributions of users over time. **Left**: A distribution which remains constant over time, following the i.i.d assumption. **Right**: Self-induced distributional shift (SIDS) results in a change in the distribution of users in our content recommendation environment (see Section 4.2 for details).

## 2 BACKGROUND

### 2.1 DISTRIBUTIONAL SHIFT AND CONTENT RECOMMENDATION

In general, **distributional shift** refers to change of the data distribution over time. In supervised learning with data $\mathbf{x}$ and labels $y$, this can be more specifically described as **dataset shift**: change in the joint distribution of $P(\mathbf{x}, y)$ between the training and test sets (Moreno-Torres et al., 2012; Quionero-Candela et al., 2009). As identified by Moreno-Torres et al. (2012), two common kinds of distributional shift are:

1. **Covariate shift**: changing $P(\mathbf{x})$. In the context of content recommendation, this corresponds to changing the user base of the recommendation system. For instance, a media outlet which publishes inflammatory content may appeal to users with extreme views while alienating more moderate users. This **self-selection effect** (Kayhan, 2015) may appear to a recommendation system as an increase in performance, leading to a feedback effect, as previously noted by Shah et al. (2018). This type of feedback effect has been identified as contributing to filter bubbles and radicalization (Pariser, 2011; Kayhan, 2015). We observe this type of change in our experiments, as shown in Figure 1.

---

[1] The authors argue that commenting on a video is a good proxy for supporting its viewpoint, since only 5 out of 900 comments they checked opposed the viewpoint of the video commmented on.

2. **Concept shift**: changing $P(y|\mathbf{x})$. In the context of content recommendation, this corresponds to changing a given user's interest in different kinds of content. For example, exposure to a fake news story has been shown to increase the perceived accuracy of (and thus presumably the interest in) the story, an example of the **illusory truth effect** (Pennycook et al., 2019).

### 2.2 META-LEARNING AND POPULATION BASED TRAINING

**Meta-learning** is the use of machine learning techniques to learn machine learning algorithms. This generally involves instantiating multiple learning scenarios which run in an **inner loop (IL)**, while an **outer loop (OL)** uses the outcomes of the inner loop(s) as data-points from which to learn which learning algorithms are most effective (Metz et al., 2019). The number of IL steps per OL step is called the **interval** of the OL.

Many recent works have focused on **multi-task meta-learning** where the OL seeks to find learning rules that generalize to unseen tasks by training the IL on a distribution of tasks - this is often used as an approach to one- or few-shot learning, e.g. Finn et al. (2017); Ren et al. (2018), or transfer learning, e.g. Andrychowicz et al. (2016). **Single-task meta-learning** includes learning an optimizer for a single task e.g. Gong et al. (2018), adaptive methods for selecting models, e.g. Kalousis (2000), or for setting hyperparameters, e.g. Snoek et al. (2012). For simplicity in this initial study we focus on single-task meta-learning.

**Population-based training** (PBT) (Jaderberg et al., 2017) is a meta-learning algorithm that trains multiple learners $L_1, ..., L_n$ in parallel, after each interval ($T$ steps of IL) applying an evolutionary OL step which consists of:

1. Evaluate the performance of each learner,
2. Replace both parameters and hyperparameters of low-performing (bottom 20%) learners with copies of those from randomly chosen high-performing (top 20%) learners (EXPLOIT),
3. Randomly perturb the hyperparameters (but not the parameters) of all learners (EXPLORE).

Two distinctive features of PBT (compared with other hyperoptimization methods, such as Bayesian optimization (Snoek et al., 2012)) are notable for us because they give the OL more control over the learning process:

1. PBT applies OL optimization to parameters, not just hyperparameters. This means the OL can directly select for parameters which lead to SIDS, instead of only being able to influence parameter values via hyperparameters, which may be much more limiting.
2. PBT uses multiple OL steps within a single training run. This gives the OL more overall influence over the dynamics and outcome of the training run.

### 2.3 SPECIFICATION AND INCENTIVES

We define **specification** as the process of a (typically human) designer instantiating a learning algorithm in a real-world *learning scenario* (see Appendix 2 for formal definitions). A **specification problem** occurs when the outcome of a learning scenario differs from the intentions of the designer. Specification is often viewed as concerned solely with the choice of performance metric, and indeed researchers often select learners solely on the basis of performance. However, our work emphasizes that the choice of learning algorithm is also an aspect of specification, as noted by Ortega & Maini (2018).

In particular, we consider this choice from the point of view **incentives**, similarly to Everitt et al. (2019). Their work focused on identifying which incentives exist, but we note that incentives may exist and yet not be pursued by a learner; for example, in supervised learning, there is an incentive to overfit the test set in order to increase test performance, but algorithms are designed to not do that. We thus distinguish between the existence of an incentive in a learner's operational context and its presence in a learner's **objective**, or **revealed specification** (Ortega & Maini, 2018), which is what a learner is "trying" to accomplish. Given an incentive that is present in the operational context, we say it is **hidden** from a learner if it does not appear in the objective, and **revealed** if it does.

## 3 SELF-INDUCED DISTRIBUTION SHIFT (SIDS) AND HIDDEN INCENTIVES FOR DISTRIBUTIONAL SHIFT (HIDS)

### 3.1 SIDS

To formally define SIDS, we assume there exists some reference data distribution, which is the distribution of data that the learner would encounter "by default". This is a standard assumption

for classification problems (Moreno-Torres et al., 2012); in reinforcement learning, the reference distribution could be the initial distribution over states, or the distribution over states which results from following some reference policy. We say that SIDS occurs whenever the behavior (e.g. actions or predictions, or mere *existence*), of the learner leads it to encounter a distribution other than this reference distribution. This definition excludes distributional shift which would happen even if the learner were not present - e.g. for a crash prediction algorithm trained on data from the summer, snowy roads in the winter are an example of distributional shift, but not *self-induced* distributional shift (SIDS).

In order to highlight the phenomenon of SIDS, we distinguish between the (often implicit) assumptions of the machine learning algorithm (e.g. the i.i.d. assumption), vs. the model of the environments in which the algorithm is trained/deployed (e.g. our synthetic content recommendation environment). This is formalized in Appendix 2. This distinction allows us to explicitly model situations in which the assumptions of a learning algorithm are violated. For instance, in Sec. 4.2 we explicitly model a partially observable environment whose underlying state determines the data distribution of the examples that a standard supervised learning algorithm observes at each time-step.

### 3.2 HIDS

Referring to Section 2.3, we say that incentives for SIDS are **hidden** if they are not part of the objective of a learner. Like SIDS, HIDS are not necessarily good or bad. Rather, our point is that designers should be cognizant of which incentives exist, and whether they are hidden or revealed to a learner. More specifically, changing the learning algorithm can reveal incentives that were previously hidden, leading learners to adopt unanticipated and potentially undesirable strategies for maximizing performance. For instance, by optimizing for performance after a *sequence* of inner loop updates, meta-learning can fail to distinguish between solving the task as intended and making the task easier via SIDS, and thus can reveal **hidden incentives for distributional shift (HIDS)**.

In many settings, such as reinforcement learning (Sutton & Barto, 1998), learners are *intended* to increase performance via SIDS. For prediction tasks, on the other hand, learners are typically not meant to seek distributional shift, *even if there is an incentive to do so*, as we illustrate with the coffee robot example in the introduction. And even in reinforcement learning, SIDS can be undesirable, as we illustrate in Sec. 4.1.

### 3.3 CONTEXT SWAPPING: A MITIGATION TECHNIQUE

We propose a technique called **context swapping** for mitigating HIDS revealed by meta-learning. The idea of context swapping is for learners to experience a "natural" distribution [2] of trajectories, $P(\tau)$, as compared to the "unnatural" distributions which can result when meta-learning is applied. Formally, we can characterize the natural distribution as:

$$P(\tau) = \int P(L)P_\mu(\tau|L)dL \qquad (1)$$

where $L$ is a learner, selected at random according to a fixed distribution $P(L)$. Here, a learner is a fully described learning algorithm[3] and $P_\mu(\tau|L)$ is the distribution over trajectories that results from running the algorithm in an environment $\mu$. Importantly, $L$ is sampled from $P(L)$, instead of being chosen via meta-learning. To provide learners with a distribution approximating $P(\tau)$, context swapping relies on training a population of $N$ learners $\{L_1, ..., L_N\}$ in parallel. Each learner inhabits one of $N$ copies $E_{1:N} \doteq \{E_1, ..., E_N\}$ of the same environment $\mu$. The $E_{1:N}$ share the same initial state distribution and time-step, but may be in different states on any particular time-step. [4]

The technique of **context swapping** consists in shuffling the learners through the different copies of the environment, so *which* copy a given learner inhabits can change at any (or every) time-step. In this work, we use a deterministic permutation of learners against environment copies, so that learner $L_i$ acts in copy $E_j$ on time-steps $t$ if and only if $j = (i + t) \mod N$. When $N$ is larger than the interval of the OL optimizer, each learner will inhabit each copy for at most a single time-step before

---

[2] Note that this concept is different from the reference distribution mentioned in 3.1; the "natural" (non-meta-learned) distribution might still exhibit SIDS, and does so in both of the environments we introduce.

[3] For example, $L$ might completely specify a deep learning algorithm including the choice of initial parameters, and $P(L)$ might be the distribution induced by the randomness of the initialization.

[4] Note that the learners consist entirely of *software*; any hardware (e.g. a robot body) would be considered part of an environment, as is typical in reinforcement learning (Sutton & Barto, 1998).

an OL step is applied. This removes the incentive for learners to manipulate the future states they encounter, although they may still have incentives to influence *each others'* future states. Under the assumption that different copies of the environment do not influence each other, this technique can address HIDS in practice, as we show in Sec. 4.1.1.

## 4 EXPERIMENTS

To clearly introduce the concepts of SIDS and HIDS, we opt for simple illustrative environments. Code for our experiments is available at `https://anonymous.4open.science/r/66c5e3a4-2a45-4d71-ae58-d097e12ebae1/`.

In Section 4.1, we introduce a "unit test" for HIDS. Our primary goal with this unit test is for the reader to walk away with a crisp understanding of HIDS. Put simply, our experiments show that you can have a learner which behaves as intended, and just by using meta-learning (e.g. PBT), and *without changing the performance metric* (e.g. loss or rewards), the learner's objective can change completely, leading to unintended behavior. On the practical side, the unit test can be used to diagnose and compare learning algorithms. We show that context swapping is an effective mitigation technique in this environment.

In Section 4.2, we model a content recommendation system. The goal of these experiments is to provide a practical understanding of different types of SIDS (concept shift and covariate shift), and to demonstrate how HIDS could create issues for real-world recommender systems. We emphasize that SIDS takes place in this environment *by construction*. The point of our experiments is that meta-learning can *increase* the rate and/or extent of SIDS. Context swapping is not effective in this environment, highlighting the need for alternatives.

### 4.1 HIDS UNIT TEST

We test whether algorithms reveal hidden incentives using a version of the prisoner's dilemma (Prisner, 2014) where an agent plays each round against its past self. The reward function is presented in Table 1. An agent in this environment has a long-term, **non-myopic** incentive for cooperation (with its future self), but a current-time-step, **myopic** incentive for defection (from its future self). The unit test evaluates whether a learning algorithm reveals the non-myopic incentive even when the agent is meant to optimize for the present reward only (i.e. uses discount rate $\gamma = 0$). Naively, we'd expect the non-myopic incentive to be *hidden* from the agent in this case, and for the agent to consistently `defect`; learning algorithms that do so pass the test. But some learning algorithms also *fail* the unit test, revealing the incentive for the agent to cooperate with its future self. While aiming for myopic behavior may seem odd, myopic learners have no incentives to cause distributional shift, since it can only improve *future* performance. And while making learners myopic is an extreme approach to managing HIDS, doing so proves to be non-trivial.

Table 1: Rewards for the HIDS unit test. Note that the myopic `defect` action always increases reward at the current time-step, but decreases reward at the next time-step - the incentive is hidden from the point of view of a myopic learner. A supposedly myopic learner 'fails' the unit test if the incentive to cooperate is revealed, i.e. if we see more `cooperate` actions than `defect`.

|  | $a_t = $ defect | $a_t = $ cooperate |
|---|---|---|
| $s_t = a_{t-1} = $ defect | $-1/2$ | $-1$ |
| $s_t = a_{t-1} = $ cooperate | $1/2$ | $0$ |

Formally, this environment is not a 2x2 game (like the original prisoner's dilemma); it is a partially observable Markov Decision Process (POMDP) (Åström, 1965; Kaelbling et al., 1998):

$$s_t = a_{t-1}$$
$$o_t = \{\}$$
$$a_t \in \{\text{defect, cooperate}\}$$
$$P(s_t, a_t) = a_t$$
$$R(s_t, a_t) = I(s_t = \text{cooperate}) + \beta I(a_t = \text{cooperate}) - 1/2$$

where $I$ is an indicator function, and $\beta = -1/2$ is a parameter controlling the alignment of incentives (see Appendix 3.1 for an exploration of different $\beta$ values.). The initial state is sampled as $s_0 \sim U(\text{defect, cooperate})$

### 4.1.1 HIDS Unit Test experimental results and discussion

We first show that agents trained with PBT fail the unit test more often when compared with baseline agents that do not use meta-learning. We use REINFORCE (Williams, 1992) with discount factor $\gamma = 0$ as the IL optimizer for these experiments. Policies are represented by a single real-valued parameter $\theta$ (initialized as $\theta \sim \mathcal{N}(0, 1)$) passed through a sigmoid whose output represents $P(a_t = \text{defect})$. PBT (with default settings, see Section 2.2) is used to tune the learning rate, with reward on the final time-step of the interval as the performance measure for PBT. We initialize the learning rate log-uniformly between $0.01$ and $1.0$ for all experiments (whether using PBT or not). We expect and confirm that the following two factors lead to higher rates of failure (cooperation):

1. **Shorter intervals:** These give the OL more opportunities to influence the population.
2. **Larger populations:** These make outliers with exceptional non-myopic performance more likely, and OL makes them likely to survive and propagate.

The baseline (no PBT) agents pass the unit test: $P(\texttt{cooperate})$ (averaged over agents) is close to 0% - see blue curves in Figure 2. However, despite the disincentive for cooperation and the myopic inner loop, agents trained with PBT and large populations fail the unit test: $P(\texttt{cooperate})$ is around 90% - see the top right subplot of Figure 2.

Furthermore, we verify that context swapping significantly mitigates the effect of HIDS, decreasing undesirable `cooperate` behaviour to near-baseline levels - see bottom rows of Figure 2. This effect can be explained as follows: Because context swapping transfers the benefits of a learner's action to the next learner to inhabit that environment, it increases that learner's fitness, and thereby reduces the *relative* fitness (as evaluated by PBT's EXPLOIT step) of the non-myopic `cooperate` behaviour. We observe some interesting exceptions with the combination of small populations and short PBT intervals. Although context swapping still significantly decreases the effect of HIDS, non-myopic `cooperate` behaviour is observed as much as 20% of the time (for #learners=10, $T = 1$; see bottom-left plot).

We also observe that PBT reveals HIDS even when $T = 1$. We provide a detailed explanation for how this might happen in Appendix 3.1.2. But we also note that for $T = 1$, the explanation that PBT operates on a longer time horizon than the inner loop does not apply, making it especially surprising that HIDS are revealed. Thus we hypothesize that there are at least 2 mechanisms by which PBT is revealing HIDS: (1) optimizing over a longer time-scale, and (2) picking up on the correlation between an agent's current policy and the underlying state. Mechanism (2) can be explained informally as reasoning as: "If I'm cooperating, then I was probably cooperating on the last time-step as well, so my reward should be higher". As support for these hypotheses, we run control experiments identifying two algorithms (each sharing only *one* of these properties) that can fail the unit test (although context swapping remains effective):

1. **Optimizing over a longer time-scale:** replacing PBT with REINFORCE as an outer-loop optimizer.The outer-loop optimizes the parameters to maximize the summed reward of the last $T$ time-steps. As with PBT, we observe non-myopic behavior, but now *only* when $T > 1$. This supports our hypothesis that the exploitation of HIDS is due not to PBT in particular, but just to the introduction of sufficiently powerful meta-learning. See Figure 2 for results.
2. **Exploiting correlation:** Q-learning with $\gamma = 0$ an $\epsilon = 0.1$-greedy behavior policy *and no meta-learning*. If either state was equally likely, the Q-values would be the average of the values in each column in Table 1, so the estimated $Q(\text{defect})$ would be larger. But the $\epsilon$-greedy policy correlates states and actions, so the top-left and bottom-right entries carry more weight in the estimates, *sometimes* causing $Q(\text{defect}) \approx Q(\text{cooperate})$ and persistent nonmyopic behavior. See Figure 3 for results, Appendix 3.1.4 for more results, and Appendix 3.1.3 for important experimental details.

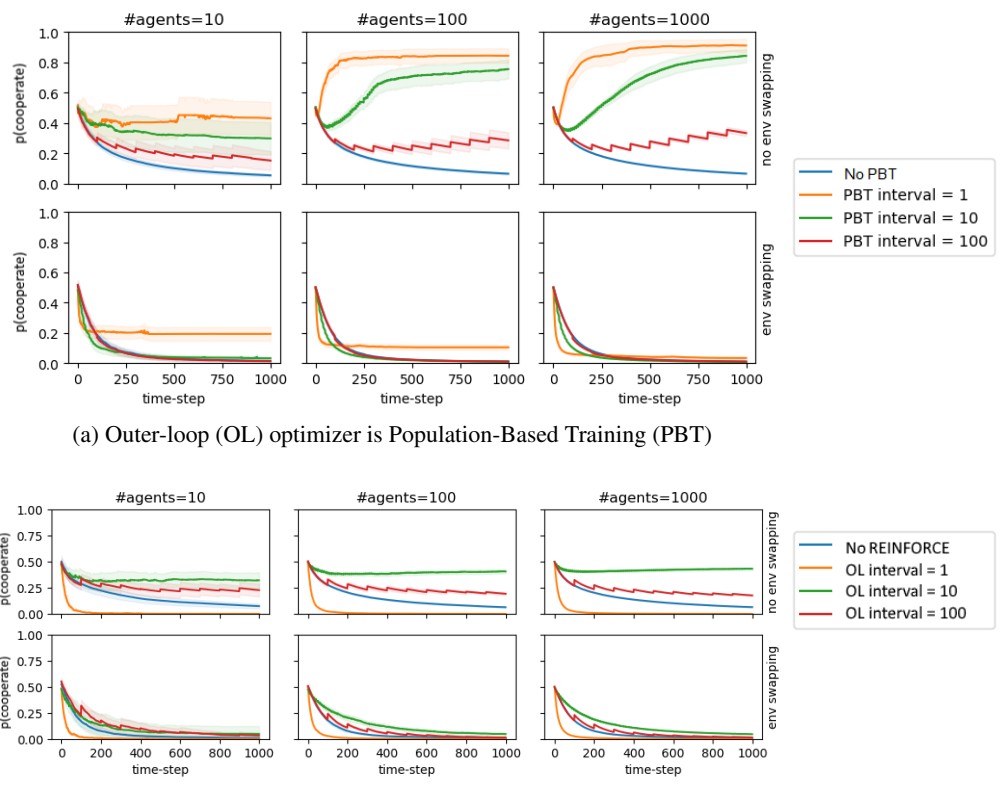

(a) Outer-loop (OL) optimizer is Population-Based Training (PBT)

(b) Outer-loop (OL) optimizer is REINFORCE

Figure 2: Average level of non-myopic `cooperate` behavior observed in the unit test for HIDS, with and without two different meta-learning algorithms (PBT and REINFORCE). Lower is better, since the goal is for non-myopic incentives to remain hidden. Despite making the inner loop fully myopic ($\gamma = 0$), both outer-loop optimizers reveal HIDS, however, leading agents to choose the `cooperate` action (**top rows of (a) and (b)**). Environment-swapping significantly mitigates HIDS (**bottom rows of (a) and (b)**).

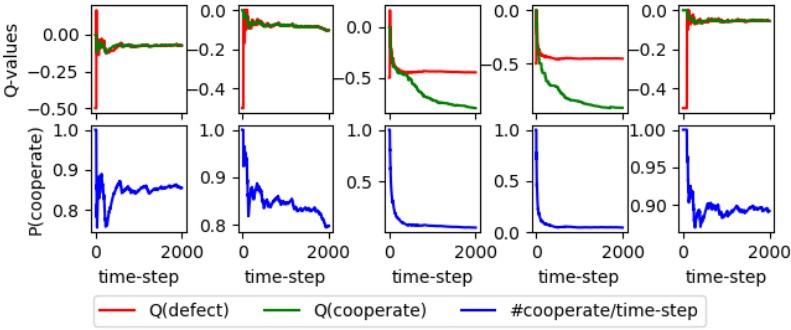

Figure 3: Q-learning sometimes fails the unit test; empirical `p(cooperate)` stays around 80-90% in 3 of 5 experiments (**bottom row**). Each column represents an independent experiment. Q-values for the `cooperate` and `defect` actions stay tightly coupled in the failure cases (**col. 1,2,5**), while in the cases passing the unit test (**col. 3,4**) the Q-value of `cooperate` is driven down over time.

## 4.2 HIDS IN CONTENT RECOMMENDATION

We now present a toy environment for modeling content recommendation of news articles, which includes the potential for SIDS by incorporating the mechanisms mentioned in Sec. 2.1, discussed as contributing factors to the problems of fake news and filter bubbles. Specifically, the environment assumes that presenting an article to a user can influence (1) their interest in similar articles, and (2) their propensity to use the recommendation service. These correspond to modeling self-induced concept shift of users, and self-induced covariate shift of the user base, respectively (see Sec. 2.1). The environment is designed to be as simple as possible while incorporating both of these effects.

This environment includes the following components, which change over (discrete) time: **User type**: $x^t$, **Article type**: $y^t$, **User interests**: $\mathbf{W}^t$ (propensity for users of each type to click on articles of each type), and **User loyalty**: $\mathbf{g}^t$ (propensity for users of each type to use the platform). At each time step $t$, a user $x^t$ is sampled from a categorical distribution, based on the loyalty of the different user types. The recommendation system (a classifier) selects which type of article to present in the top position, and finally the user 'clicks' an article $y^t$, according to their interests.

User loyalty for user type $x^t$ undergoes **covariate shift**: in accordance with the **self-selection effect**, $g^t$ increases or decreases proportionally to that user type's interest in the top article. The interests of user type $x^t$ (represented by a column of $\mathbf{W}^t$) also change, undergoing **concept shift**; in accordance with the **illusory truth effect**, their interest in the topic of the top article (as chosen by the recommender system) always increases. The update rates of $g^t$, $\mathbf{W}^t$ are specified by $\alpha_1, \alpha_2$.

Formally, this environment is similar to a POMDP\R, i.e. a POMDP with no reward function, also known as a **world model** (Armstrong & O'Rourke, 2017; Hadfield-Menell et al., 2017); the difference is that the learner observes the input ($o_{\text{pre}}^t$) before acting and only observes the target ($o_{\text{post}}^t$) after acting. The states, observations, and actions given below. For further details on this environment, including the state transition function, see Appendix 3.2.1.

$$s^t = (\mathbf{g}^t, \mathbf{W}^t, x^t, y^t)$$
$$o_{\text{pre}}^t, \ a^t, \ o_{\text{post}}^t = (x^t, \hat{y}^t, y^t)$$

### 4.2.1 CONTENT RECOMMENDATION EXPERIMENTAL RESULTS AND DISCUSSION

Our recommender system is a 1-layer MLP trained with SGD-momentum. Actions are sampled from the MLP's predictive distribution. For PBT, we use $T = 10$ and 20 agents, and use accuracy to evaluate performance. We run 20 trials, and match random seeds for trials with and without PBT. See Appendix 3.2.2 for full experimental details.

We find that PBT yields significant improvements in training time and accuracy, but also greater distributional shift; see Figure 4. User base and user interests both change faster with PBT, and in particular user interests change more overall. We observe that the distributions over user types typically saturate (to a single user type) after a few hundred time-steps (Figure 1; Figure 4, Right). We run long enough to reach such states, to demonstrate that the increase in SIDS from PBT is not transitory. The environment has a number of free parameters, and our results are qualitatively consistent so long as (1) the initial user distribution is approximately uniform, and (2) the covariate shift rate ($\alpha_1$) is faster than the concept shift rate ($\alpha_2$). See Appendix 3.2.4 for details.

We measure concept shift (change in $P(y|\mathbf{x})$) as the cosine distance between each user types' initial and current interest vectors. And we measure covariate shift (change in $P(\mathbf{x})$) as the KL-divergence between the current and initial user distributions, parametrized by $\mathbf{g}^1$ and $\mathbf{g}^t$, respectively. In Figure 5, we plot concept shift and covariate shift as a function of accuracy. We observe that for both types of SIDS, at low levels of accuracy PBT actually causes *less* shift than occur in baseline agents; HIDS are only observed for accuracies above 60%. This suggests that only relatively strong performers are able to pick up on the HIDS revealed by PBT. See Figure 5.

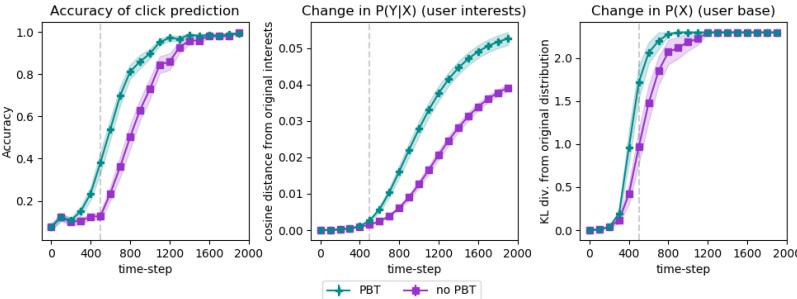

Figure 4: Content recommendation experiments. **Left**: using Population Based Training (PBT) increases accuracy of predictions faster, leads to a faster and larger drift in users' interests, $P(y|\mathbf{x})$, (**Center**); as well as the distribution of users, $P(\mathbf{x})$, (**Right**). Shading shows std error over 20 runs.

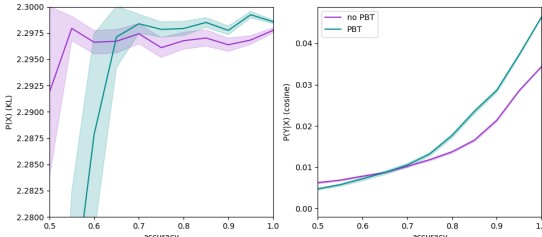

Figure 5: Amount of self-induced covariate shift (**left**) and self-induced concept shift (**right**) as a function of performance (accuracy) averaged over all trials, learners, and time-steps. Only relatively strong learners (those which achieve accuracy > 60%) exhibit HIDS.

## 5 RELATED WORK

**SIDS in practice:** We introduce the term SIDS, but we are far from the first to study such problems. Caruana et al. (2015) provide an example of asthmatic patients having lower predicted risk of pneumonia. Treating asthmatics with pneumonia less aggressively on this basis would be an example of harmful SIDS; the *reason* they had lower pneumonia risk was because they had received *more* aggressive lung-related care already. Schulam & Saria (2017) note that such predictive models are commonly used to inform decision-making, and propose modeling counterfactuals (e.g. "how would this patient fare with less aggressive treatment") to avoid making such (potentially) self-refuting predictions. While their goal is to make accurate predictions in the presence of SIDS, our goal is to identify and manage *incentives for* SIDS. Environments with agents that that react to a learner (such as adversaries) naturally produce SIDS. Goodfellow (2019) argues that adversarial defenses that do not account for distributional shift are critically flawed.

**Non-i.i.d bandits:** Contextual bandits (Wang et al., 2005; Langford & Zhang, 2008) are frequently discussed as an approach to content recommendation (Li et al., 2010). While bandit algorithms typically make the i.i.d. assumption, counter-examples exist (Gheshlaghi Azar et al., 2014; Shah et al., 2018); most famously, adversarial bandits (Auer et al., 1995). Closest to our work is Shah et al. (2018), who consider self-induced covariate shift in the context of multi-armed bandits. Our task in Sec. 4.2 is similar to their problem statement, but more general in that we include user features, thus disentangling covariate shift and concept shift. Our motivation is also different: Shah et al. (2018) seek to exploit SIDS, whereas we aim to avoid hidden incentives for SIDS.

**Safety and incentives:** Understanding and managing the incentives of learners is also a focus of Armstrong & O'Rourke (2017); Everitt (2018); Everitt et al. (2019); Cohen et al. (2019). Emergent incentives to influence the world (such as HIDS) are at the heart of many concerns about the safety of advanced AI systems (Omohundro, 2008; Bostrom, 2014). Yet it is unclear if or when machine learning systems might pursue such "instrumental goals" in practice. Indeed, Drexler (2019) argues that machine learning should and typically does use time- and resource-bounded problem statements, making dangerous instrumental goals less likely to emerge. The same idea underlies

several more concrete approaches to building safe superintelligent AI systems: myopic reinforcement learning (Leike et al., 2018; Knox & Stone, 2008; Cohen et al., 2019) and its application in iterated amplification (Christiano et al., 2018; Cotra, 2017) and question answering systems (Everitt et al., 2019; Armstrong & O'Rorke, 2017). Managing HIDS seems critically important for the safety of these approaches: they rely on enforcing myopia, which our experiments show is not straightforward.

**HIDS and meta-learning:** As far as we know, our work is the first to consider the problem of HIDS, or its relation to meta-learning. A few previous works have some relevance or resemblance. Rabinowitz (2019) documents qualitative differences in learning behavior when meta-learning is applied. MacKay et al. (2019) and Lorraine & Duvenaud (2018) view meta-learning as a bilevel optimization problem, with the inner loop playing a best-response to the outer loop. In our work, the inner loop is unable to achieve such best-response behavior; the outer loop is too powerful (see Fig. 2). Finally, Sutton et al. (2007) note that meta-learning can change learning behavior in a way that improves performance by preventing convergence of the inner loop. Their goal of improving performance by "tracking" local characteristics of the environment is orthogonal to our goal of managing incentives to control such local characteristics.

## 6 Discussion and Conclusion

We have identified the phenomenon of self-induced distributional shift (SIDS), and the problems that can arise when there are hidden incentives for algorithms to induce distributional shift (HIDS). Our work highlights the interdisciplinary nature of issues with real-world deployment of ML systems - we show how HIDS could play a role in important technosocial issues like filter bubbles and the propagation of fake news. There are a number of potential implications for our work:

1. When HIDS are a concern, our methodology and environments can be used to help diagnose whether and to what extent the final performance/behavior of a learner is due to SIDS and/or incentives for SIDS, i.e. to quantify their influence on that learner.
2. Comparing this quantitative analysis for different algorithms could help us understand which features of algorithms affect their propensity to reveal HIDS, and aid in the development of safer and more robust algorithms.
3. Characterizing and identifying HIDS in these tests is a first step to analyzing and mitigating other (problematic) incentives, as well as to developing theoretical understanding of incentives.

Broadly speaking, our work emphasizes that the choice of machine learning algorithm plays an important role in specification, independently of the choice of performance metric. A learner can use SIDS to increase performance *according to the intended performance metric*, and yet still behave in an undesirable way, if we did not intend the learner to improve performance by that *method*. In other words, performance metrics are incomplete specifications: they only specify our goals or *ends*, while our choice of learning algorithm plays a role in specifying the *means* by which we intend an learner to achieve those ends. With increasing deployment of ML algorithms in daily life, we believe that (1) understanding incentives and (2) specifying desired/allowed means of improving performance are important avenues of future work to ensure fair, robust, and safe outcomes.

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

APPENDICES

# 1 CONTENT RECOMMENDATION IN THE WILD

Filter bubbles, the spread of fake news, and other techno-social issues are widely reported to be responsible for the rise of populism (Groshek & Koc-Michalska, 2017), increase in racism and prejudice against immigrants and refugees (Noble, 2018), increase in social isolation and suicide (Luxton et al., 2012), and, particularly with reference to the 2016 US elections, are decried as threatening the foundations of democracy (El-Bermawy, 2016). Even in 2013, well before the 2016 American elections, a World Economic Forum report identified these problems as a global crisis (Lee Howell, 2013).

We focus on two related issues in which content recommendation algorithms play a role: fake news and filter bubbles.

## 1.1 FAKE NEWS

Fake news (also called false news or junk news) is an extreme version of yellow journalism, propaganda, or clickbait, in which media that is ostensibly providing information focuses on being eye-catching or appealing, at the expense of the quality of research and exposition of factual information. Fake news is distinguished by being specifically and deliberately created to spread falsehoods or misinformation (Merriam-Webster, 2017; Mihailidis & Viotty, 2017).

Why does fake news spread? It may at first seem the solution is simply to educate people about the truth, but research tells us the problem is more multifaceted and insidious, due to a combination of related biases and cognitive effects including **confirmation bias** (people are more likely to believe things that fit with their existing beliefs), **priming** (exposure to information unconsciously influences the processing of subsequent information, i.e. seeing something in a credible context makes things seem more credible) and the **illusory truth effect** (i.e. people are more likely to believe something simply if they are told it is true).

Allcott & Gentzkow (2017) track about 150 fake news stories during the 2016 US election, and find the average American adult saw 1-2 fake news stories, just over half believed the story was true, and likelihood of believing fake news increased with ideological segregation (polarization) of their social media. Shao et al. (2018) examine the role of social bots in spreading fake news by analyzing 14 million Twitter messages. They find that bots are far more likely than humans to spread misinformation, and that success of a fake news story (in terms of human retweets) was heavily dependent on whether bots had shared the story.

Pennycook et al. (2019) examine the role of the illusory truth effect in fake news. They find that even a single exposure to a news story makes people more likely to believe that it is true, and repeat viewings increase this likelihood. They find that this is not true for extremely implausible statements (e.g. "the world is a perfect cube"), but that "only a small degree of potential plausibility is sufficient for repetition to increase perceived accuracy" of the story. The situation is further complicated by peoples' inability to distinguish promoted content from real news - Amazeen & Wojdynski (2018) find that fewer than 1/10 people were able to tell when content was an advertisement, even when it was explicitly labelled as such. Similarly, Fazio et al. (2015) find that repeated exposure to incorrect trivia make people more likely to believe it, even when they are later able to identify the trivia as incorrect.

## 1.2 FILTER BUBBLES

Filter bubbles, a term coined and popularized by Pariser (2011) are created by positive or negative feedback loops which encourage users or groups of users towards increasing within-group similarity, while driving up between-group dissimilarity. The curation of this echo chamber is called **self-selection** (people are more likely to look for or select things that fit their existing preferences), and favours what Techopedia (2018) calls intellectual isolation. In the context of social and political opinions, this is often called the **polarization effect** (Wikipedia contributors, 2018).

Filter bubbles can be encouraged by algorithms in two main ways. The first is the most commonly described: simply by showing content that is similar to what a user has already searched for, search or recommender systems create a positive feedback loop of increasingly-similar content (Pariser, 2011; Kayhan, 2015). The second way is similar but opposite - if the predictions of an algorithm are good for a certain group of people, but bad for others, the algorithm can do better on its metrics by driving hard-to-predict users away. Then new users to the site will either be turned off entirely, or see an artificially homogenous community of like-minded peers, a phenomena Shah et al. (2018) call **positive externalities**.

In a study of 50,000 US-based internet users, Flaxman & Goel (2015) find that two things increase with social media and search engine use: (1) exposure of an individual to opposing or different viewpoints, and (2) mean ideological distance between users. Many studies cite the first result as evidence of the *benefits* of internet and social media (Robson, 2018; Bakshy et al., 2015), but the correlation of exposure with ideological distances demonstrates that exposure is not enough, and might even be counterproductive.

Facebook's own study on filter bubbles results show that the impact of the news feed algorithm on filter bubble "size" (a measure of homogeneity of posts relative to a baseline) is almost as large as the impact of friend group composition (Bakshy et al., 2015). Kayhan (2015) specifically study the role of search engines in confirmation bias, and find that search context and the similarity of results in search engine results both reinforce existing biases and increase the likelihood of future biased searches. Nguyen et al. (2014) similarly study the effect of recommender systems on individual users' content diversity, and find that the set of options recommended narrows over time.

Filter bubbles create an ideal environment for the spread of fake news: they increase the likelihood of repeat viewings of similar content, and because of the illusory truth effect, that content is more likely to be believed and shared (Pennycook et al., 2019; DiFranzo & Gloria-Garcia, 2017; Pariser, 2011). We are not claiming that HIDS are entirely or even mostly responsible for these problems, but we do note that they can play a role that is worth addressing.

## 2 FORMAL DEFINITIONS

Here we formalize the concepts of learning scenario, operational context, and problem statement for maximal clarity and to highlight issues of applying machine learning in practice. A **learning scenario**, $\mathfrak{L}$ consists of a learning algorithm, $L$, operational context, $\varpi$, and performance metric, $\mathcal{L}$. The **performance metric** is a quantification of learners' performance, used during testing, validation, and sometimes training, to evaluate and compare learners. The **operational context** is the real-world setting where a learner operates. It includes things like the training data or reinforcement learning environment. Significantly, it also includes external factors, such as human users and the computer hardware running the learning algorithm, which may potentially influence the data or states the learner encounters. The **learning algorithm** instantiates a (potentially stochastic) mapping:

$$L : (\varpi, \mathcal{L}) \longrightarrow O \tag{2}$$

where $O$ is the output of learning, including things like a learned model, learning curves, and/or a complete log of any computations or data processed during learning.

A **problem statement**, $\tilde{\varpi}$, is a model of an operational context, used by humans to analyze properties of learning scenarios. Like all models, problem statements make simplifying assumptions, such as assuming i.i.d. inputs in the presence of distributional shift. Researchers often design a learning algorithm with a specific problem statement, $\tilde{\varpi}^{\text{intended}}$, in mind, and only evaluate it in operational contexts that are carefully controlled to match $\tilde{\varpi}^{\text{intended}}$. On the other hand, practitioners often deploy learning algorithms in less controlled operational contexts that are not faithful to $\tilde{\varpi}^{\text{intended}}$. Our work employs and advocates for an empirical methodology of testing a learning algorithm designed for $\tilde{\varpi}^{\text{intended}}$ in a problem statement $\tilde{\varpi}^{\text{realistic}}$ which seems more realistic (but may still fail to capture important aspects of an operational context), in order to detect possible failure modes and develop mitigation strategies.

## 3 EXTRA EXPERIMENTS AND REPRODUCIBILITY DETAILS

### 3.1 HIDS UNIT TEST

#### 3.1.1 ALIGNMENT OF INCENTIVES EXPLORATION

This section presents an exploration of the parameter $\beta$, which controls the alignment of incentives in the HIDS unit test.

To clarify the interpretation of experiments, we distinguish between environments in which myopic (defect) vs. nonmyopic (cooperate) incentives are **opposed**, **orthogonal**, or **compatible**. Note that in this unit test myopic behaviour (defection) is what we want to see.

1. **Incentive-opposed**: Optimal myopic behavior is incompatible with optimal nonmyopic behavior (classic prisoner's dilemma; these experiments are in the main paper).
2. **Incentive-orthogonal**: Optimal myopic behavior may or may not be optimal nonmyopic behavior.
3. **Incentive-compatible**: Optimal myopic behavior is necessarily also optimal nonmyopic behavior.

Table 2: $\beta$ controls the extent to which myopic and nonmyopic incentives are aligned.

| $\beta$ | Environment | Cooperating |
|---|---|---|
| $< 0$ | incentive-opposed | yields less reward on the current time-step (myopically detrimental) |
| $= 0$ | incentive-orthogonal | does not affect the current reward (myopically indifferent) |
| $> 0$ | incentive-compatible | yields more reward on the current time-step (myopically beneficial) |

We focused on incentive-opposed environment ($\beta = -1/2$) in the main paper in order to demonstrate that HIDS can be powerful enough to change the behavior of the system in an undesirable way. Here we also explore incentive-compatible and incentive-orthogonal environments because they provide useful baselines, helping us distinguish a systematic bias towards nonmyopic behavior from other reasons (such as randomness or optimization issues) for behavior that does not follow a myopically optimal policy.

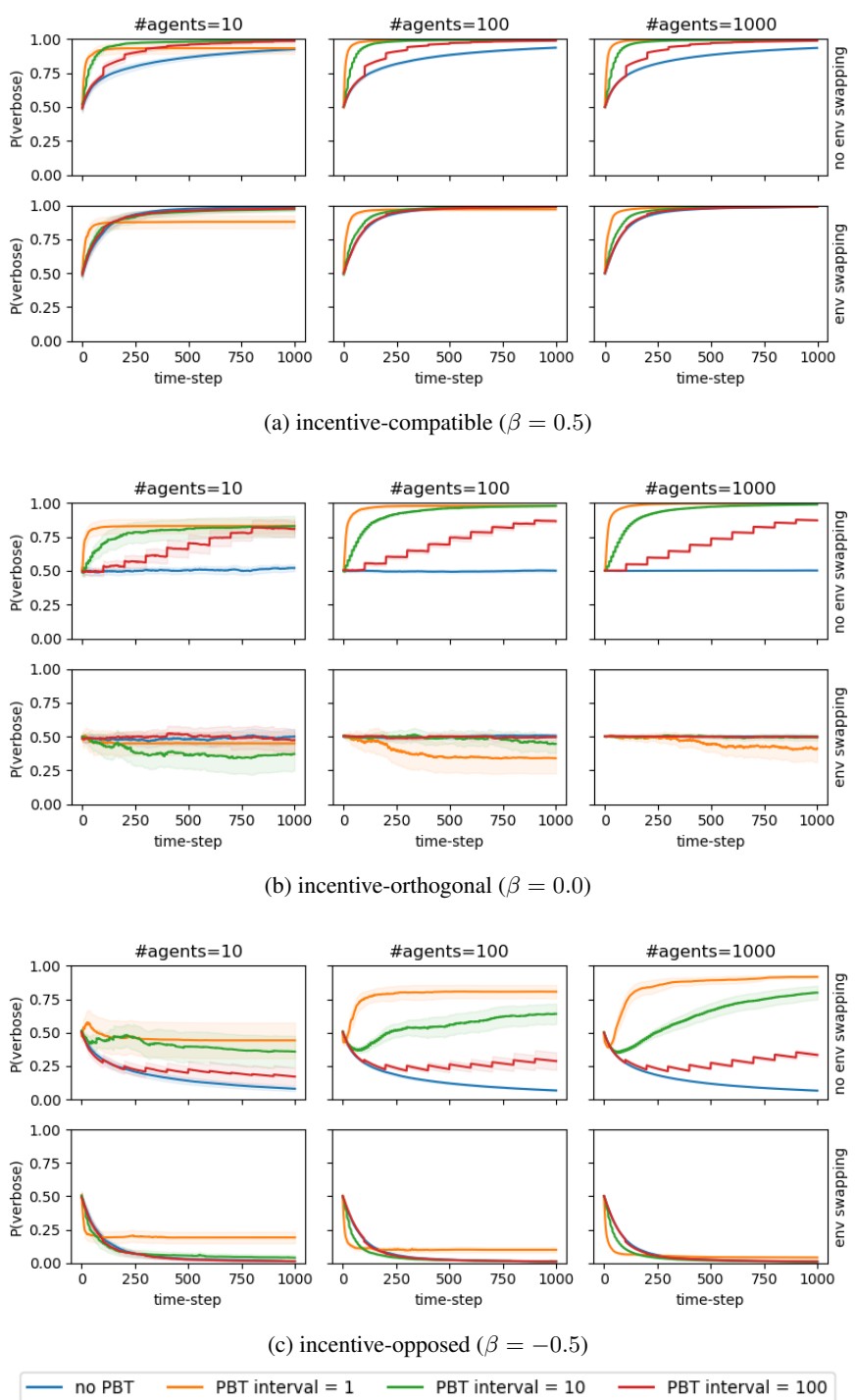

Figure 6: Average level of non-myopic (i.e. `cooperate`) behavior learned by agents in the unit test for HIDS. Despite making the inner loop fully myopic ($\gamma = 0$), population-based training (PBT) can cause HIDS, leading agents to choose the `cooperate` action (**top row**). Environment-swapping successfully prevents this (**bottom row**). Columns (from left to right) show results for populations of 10, 100, and 1000 learners. In the legend, "interval" refers to the interval ($T$) of PBT (see Sec. 2.2). Sufficiently large populations and short intervals are necessary for PBT to induce nonmyopic behavior.

### 3.1.2 WORKING THROUGH A DETAILED EXAMPLE FOR PBT WITH $T = 1$

To help provide intuition on how (mechanistically) PBT could lead to persistent levels of cooperation, we walk through a simple example (with no inner loop). Consider PBT with $T = 1$ and a population of 5 deterministic agents $A_1, ..., A_5$ playing cooperate and receiving reward of $r(A_i) = 0$. Now suppose $A_1$ suddenly switches to play defect. Then $r(A_1) = 1/2$ on the next time-step (while the other agents' reward is still 0), and so PBT's EXPLOIT step will copy $A_1$ (without loss of generality to $A_2$). On the following time-step, $r(A_2) = 1/2$, and $r(A_1) = -1/2$, so PBT will clone $A_2$ to $A_1$, and the cycle repeats. Similar reasoning applies for larger populations, and $T > 1$.

### 3.1.3 Q-LEARNING EXPERIMENT DETAILS

We show that, under certain conditions, Q-learning can learn to (primarily) cooperate, and thus fails the HIDS unit test. We estimate Q-values using the sample-average method, which is guaranteed to converge in the fully observed, tabular case (Sutton & Barto, 1998). The agent follows the $\epsilon$-greedy policy with $\epsilon = 0.1$. In order to achieve this result, we additionally start the agent off with one synthetic memory where both state and action are defect and therefor $R(\text{defect}) = -.5$, and we hard-code the starting state to be cooperate (which normally only happens 50% of the time). Without this kind of an initialization, the agent always learns to defect. However, under these conditions, we find that 10/30 agents learned to play cooperate most of the time, with $Q(\text{cooperate})$ and $Q(\text{defect})$ both hovering around $-0.07$, while others learn to always defect, with $Q(\text{cooperate}) \approx -0.92$ and $Q(\text{defect}) \approx -0.45$. context swapping, however, prevents majority-cooperate behavior from ever emerging, see Figure 9.

### 3.1.4 Q-LEARNING: FURTHER RESULTS

To give a more representative picture of how often Q-learning fails the unit test, we run a larger set of experiments with Q-learning, results are in Figure 8. It's possible that the failure of Q-learning is not persistent, since we have not proved otherwise, but we did run much longer experiments and still observe persistent failure, see Figure 7.

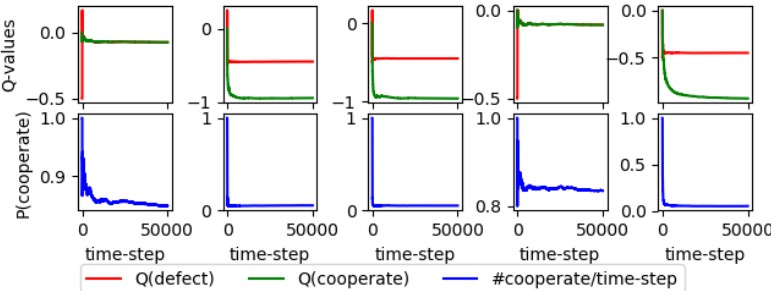

Figure 7: The same experiments as Figures 3, 8, run for 50,000 time-steps instead of 3000, to illustrate the persistence of non-myopic behavior.

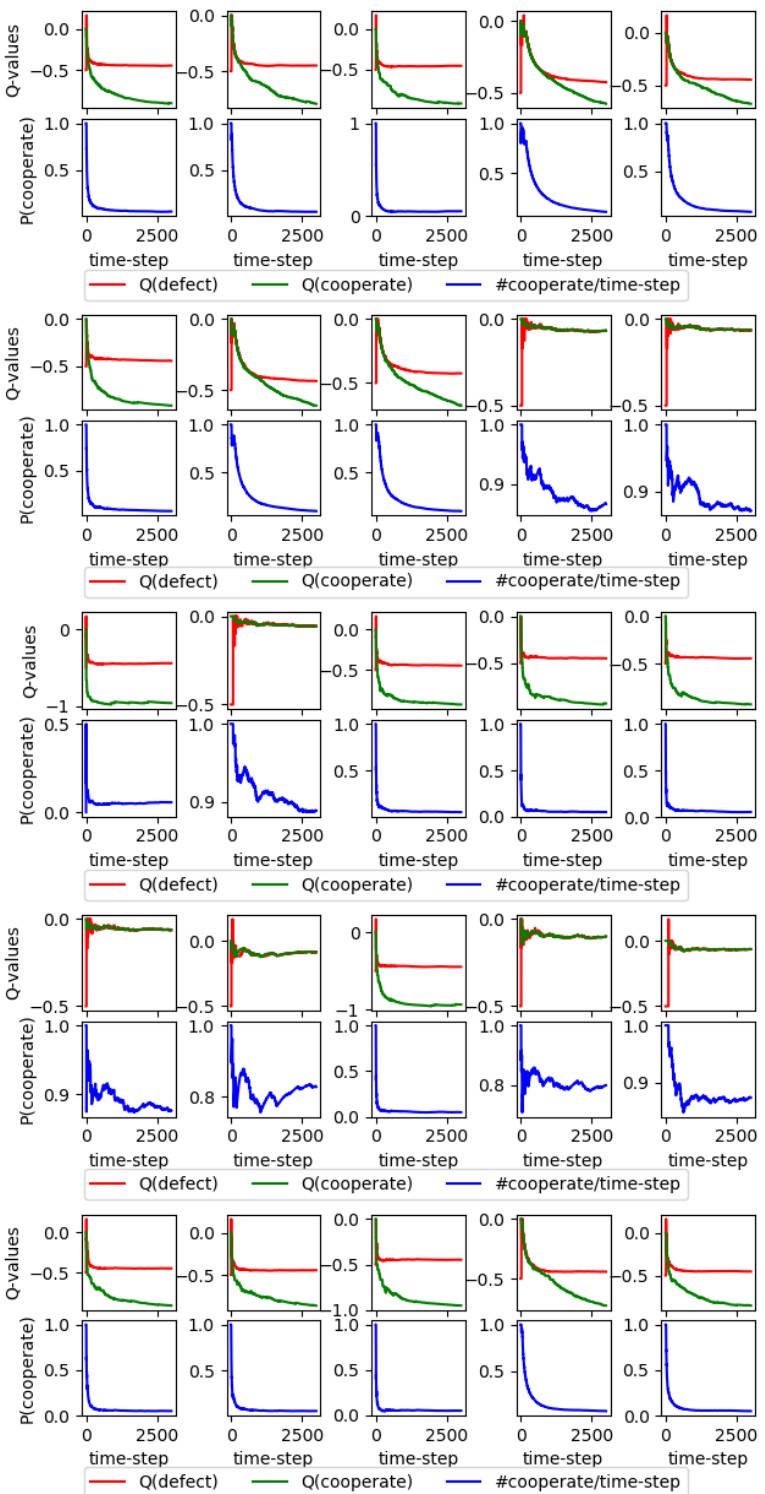

Figure 8: More independent experiments with Q-learning, exactly following Figure 3. Q-learning fails the unit test in a total of 10/30 experiments (including those from Figure 3).

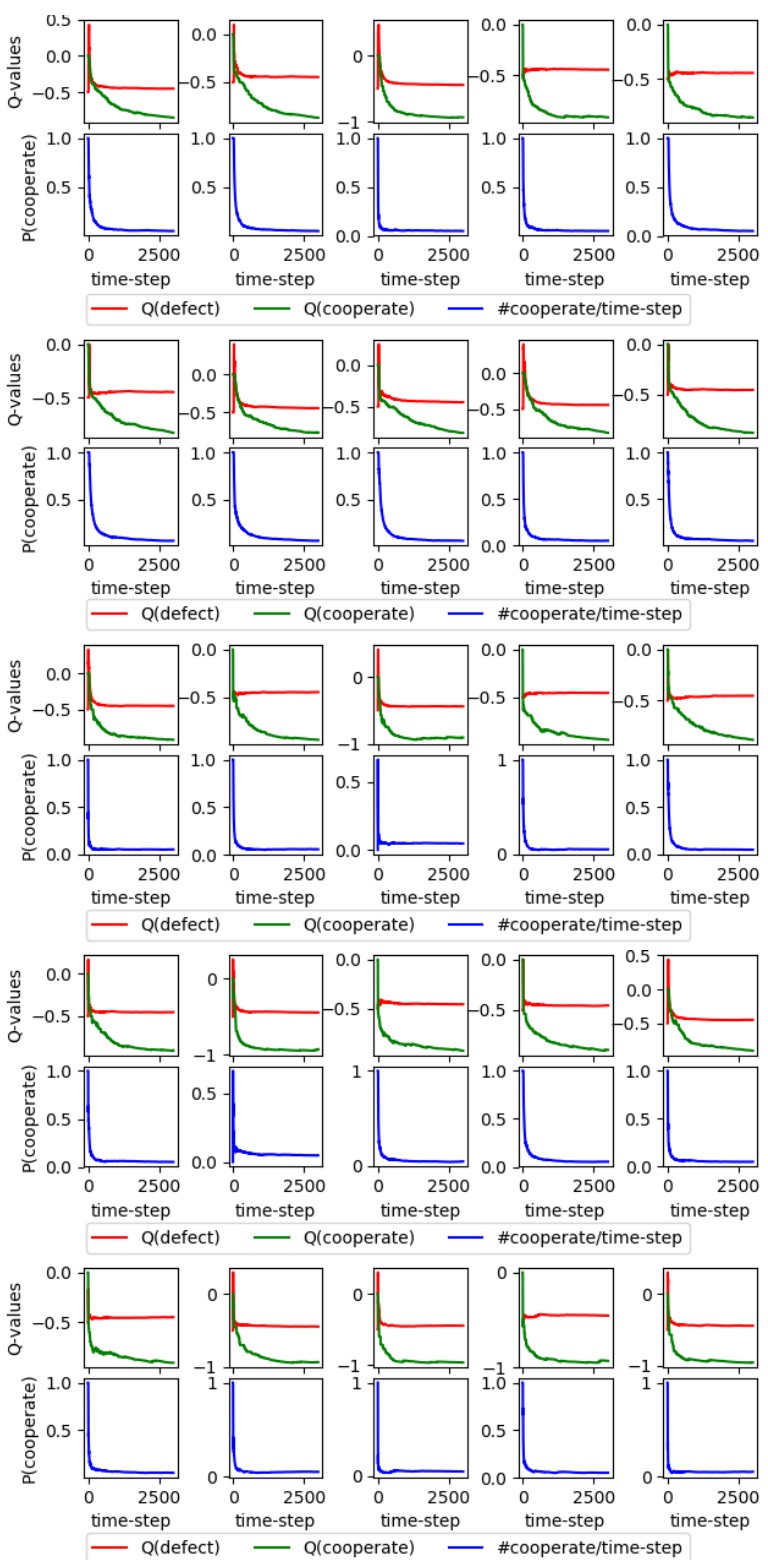

Figure 9: More independent experiments with Q-learning, exactly following Figure 3, except also using context swapping. This leads to a 100% success rate on the unit test.

## 3.2 CONTENT RECOMMENDATION

### 3.2.1 ENVIRONMENT DETAILS

The evironment has the following components:

1. **User type**, $x^t$: categorical variable representing different types of users. The content recommender conditions its predictions on the type of the current user.

2. **User loyalty**, $\mathbf{g}^t$: the propensity for users of each type to use the platform. User $x^t$ is sampled from a categorical distribution with parameters given by $\text{softmax}(\mathbf{g}^t)$.

3. **Article type**, $y^t$: a categorical variable (one-hot encoding) representing the type of article selected by the user.

4. **User interests**, $\mathbf{W}^t$: a matrix whose entries $W_{x,y}^t$ represent the average interest user of type $x$ have in articles of type $y$.

At each time step $t$, a user $x^t$ is sampled from a categorical distribution (based on the loyalty of the different user types), then the recommendation system selects which type of article to present in the top position, and finally, the user selects an article. The goal of the recommendation system is to predict the likelihood that the user would click on each of the available articles, in order to select the one which is most interesting to the user.

User loyalty for $x^t$ then changes in accordance with the self-selection effect, increasing or decreasing proportionally to their interest in the top article. The interests of user type $x^t$ (represented by a column of $\mathbf{W}^t$) also change; in accordance with the illusory truth effect, their interest in the topic of the top article (as chosen by the recommender system) always increases. Overall, this environment is an extremely crude representation of reality, but it allows us to incorporate both the effects of self-selection (via covariate shift), and the illusory truth effect (via concept shift).

Formally, this environment is similar to a POMDP\R, i.e. a POMDP with no reward function, also known as a **world model** (Armstrong & O'Rourke, 2017; Hadfield-Menell et al., 2017); the difference is that the learner observes the input before acting and only observes the target after acting. The states, observations, and actions given below.

$$s^t = (\mathbf{g}^t, \mathbf{W}^t, x^t, y^t)$$
$$o_{\text{pre}}^t,\ a^t,\ o_{\text{post}}^t = (x^t, \hat{y}^t, y^t)$$

The state transition function is defined by:

$$\mathbf{g}_{x^t}^{t+1} = \mathbf{g}_{x^t}^t + \alpha_1 W_{x^t, \hat{y}^t}^t$$

$$\mathbf{W}_{x^t, \hat{y}^t}^{t+1/2} = W_{x^t, \hat{y}^t}^t + \alpha_2; \quad \mathbf{W}_{x^t}^{t+1} = \frac{\mathbf{W}_{x^t}^{t+1/2}}{\|\mathbf{W}_{x^t}^{t+1/2}\|_2}$$

$$x^{t+1} \sim \text{softmax}(\mathbf{g}^{t+1})$$

$$y^{t+1} \sim \text{softmax}(\mathbf{W}_{x^{t+1}}^{t+1})$$

Where $\hat{y}^t$ is the top article as chosen by the recommender, and $\alpha_1$, $\alpha_2$ represent the rate of covariate and concept shift (respectively). The update for $\mathbf{W}^{t+1}$ merely increases the interest of user type $x^t$ in article type $\hat{y}^t$, then normalizes the interests for that user type.

### 3.2.2 REPRODUCIBILITY DETAILS

For these experiments, the recommendation system is a ReLU-MLP with 1 hidden layer of 100 units, trained via supervised learning with SGD (learning rate = 0.01) to predict which article a user will select. Actions are sampled from the MLP's predictive distribution. We apply PBT without any hyperparameter selection (this amounts to just doing the EXPLOIT step), and an interval of 10, selecting on accuracy. We use a population of 20 learners (whether applying PBT or not), and match random seeds for the trials with and without PBT. We initialize $\mathbf{g}^1$ and $\mathbf{W}^1$ to be the same across the 20 copies of the environment (i.e. the learners start with the same user population), but these values diverge throughout learning. For the environment, we set the number of user and article types both to 10. Initial user loyalties are randomly sampled from $\mathcal{N}(0, 0.03)$, $\alpha_1 = 0.03$, and $\alpha_2 = 0.003$.

### 3.2.3 CONTEXT SWAPPING IN CONTENT RECOMMENDATION

We believe context swapping is not appropriate for the content recommendation environment, since when the environments diverge, optimal behavior may differ across environments. Nevertheless, we ran experiments with it for completeness. The main effect appears to be to hamper learning when PBT is not used, see Figure 10. Notably, it does not appear to significantly influence the rate or extent of SIDS when combined with PBT.

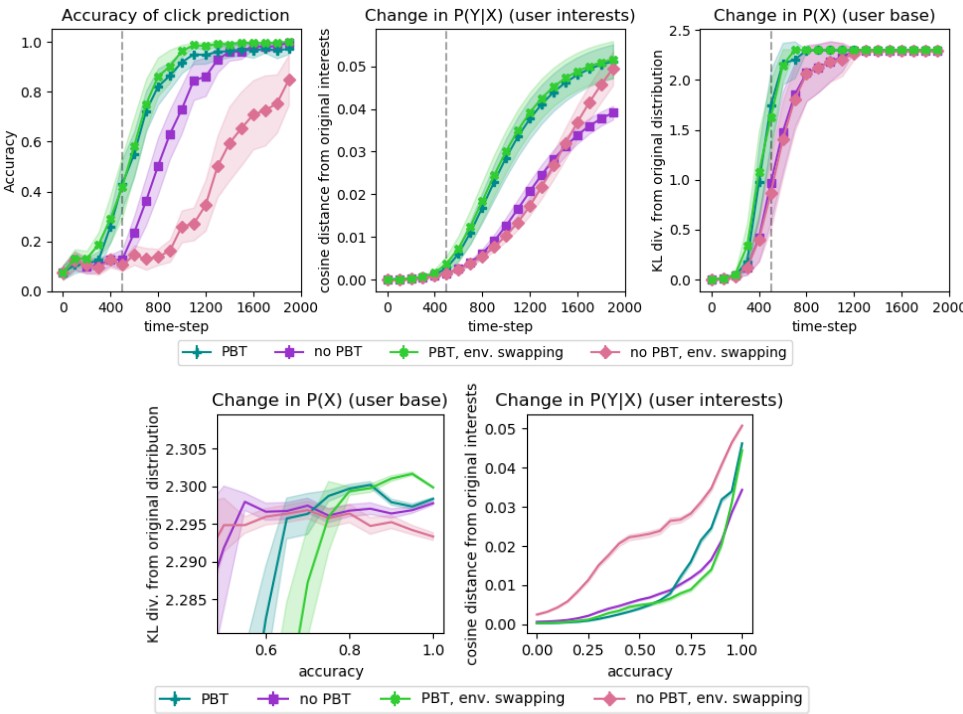

Figure 10: context swapping doesn't have the desired effect in the content recommendation environment.

### 3.2.4 EXPLORATION OF ENVIRONMENT PARAMETERS

In Figure 11, we examine the effect of the rate-of-change parameters ($\alpha_1$, $\alpha_2$) of the content recommendation environment on the results provided in the paper. As noted there, our results are qualitatively consistent so long as (1) the initial user distribution is approximately uniform, and (2) the covariate shift rate ($\alpha_1$) is faster than the concept shift rate ($\alpha_2$). These distributions are updated by different mechanisms, and are not directly comparable. Concept shift changes the task more radically, requiring a learner to change its predictions, rather than just become accurate on a wider range of inputs. We conjecture that changes in $P(y|x)$ must therefore be kept smooth enough for the outer loop to have pressure to capitalize on HIDS.

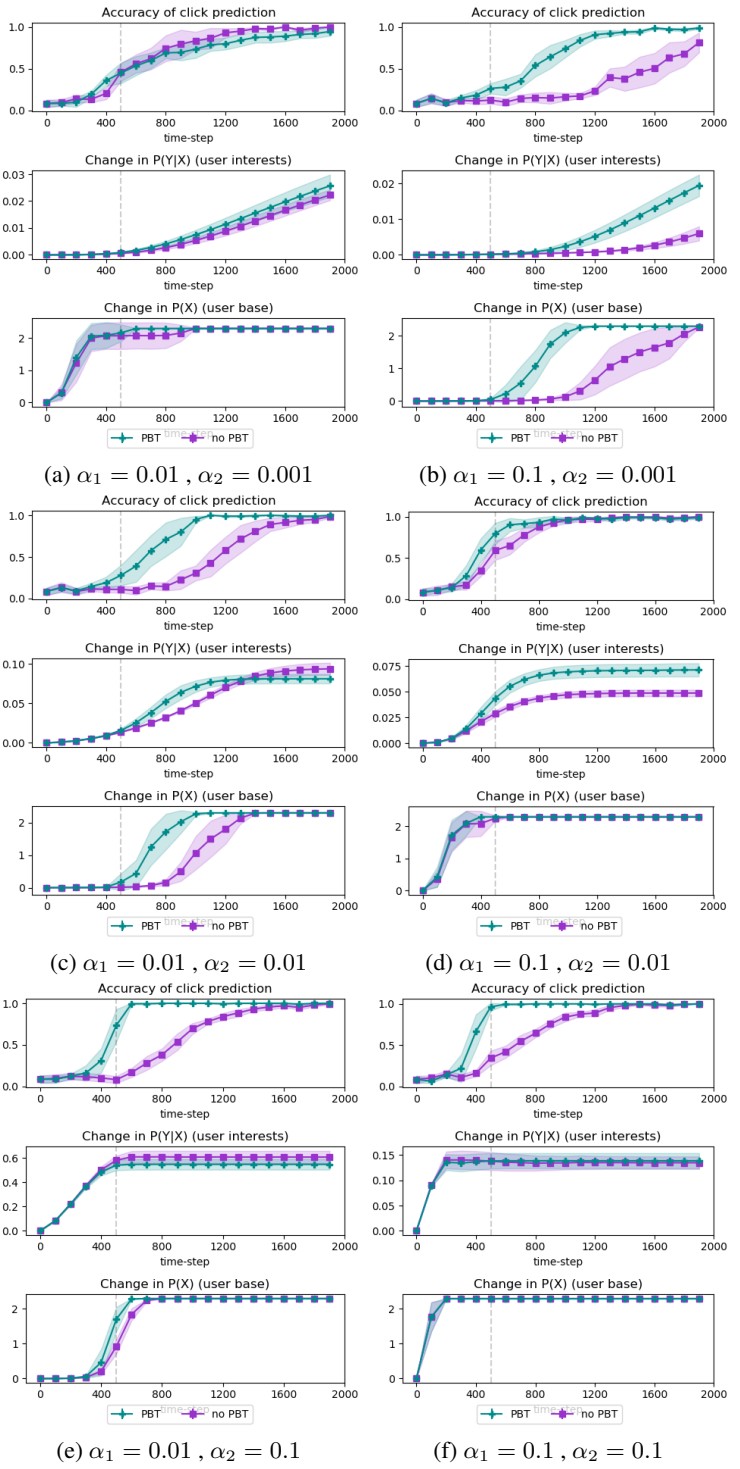

Figure 11: Content recommendation results for different values of $\alpha_1$, $\alpha_2$.

