# OpenReview forum: "Hidden incentives for self-induced distributional shift"
_ICLR.cc/2020/Conference — Reject_

### Official Review · AnonReviewer2 · 2019-10-23
**Official Blind Review #2**

**Rating:** 6

**Review:**

The authors study the phenomena of self-introduced distributional shift. They define the term along with the term hidden incentives for distributional shift. The latter describes factors that motivate the learner to change the distribution in order to achieve a higher performance. The authors study both phenomena in two domains (one being a prisoner dilemma and the other a recommender system) and show how meta-learning reveals the hidden incentives for distributional shift. They then propose an approach based on swapping learners between environments to reduce self introduced distributional shift.

In my opinion this paper should be (weak) rejected for several reasons.

-	The paper is poorly written. Some sentences are hard to comprehend, even after repeated reading. It feels hastily written and should be carefully proofread with the help of a proficient English speaker.
-	The very first paragraph is not a helpful example. While the authors provide an example that that illustrates a distributional shift, they don’t describe what the shift is and why that shift in a negative consequence for learning. Therefore, the example is not helpful, but rather confusing.
-	In point 3 of the contributions, the authors state ‘that meta-learning reveals HIDS in these environments’. Is this true for all meta-learning approaches? The current statement overclaims their findings. If the authors decide to keep it, they should provide a description of the types of meta-learning approaches that reveal HIDS and which don’t or evaluate all different meta-learning approaches.
-	Section 4.1 is hard to follow. The authors state that the meta-learner is used to tune the learning rate – they fail to clearly explain how exactly the learning rate tuning results in a non-myopic behavior for a myopic learner.
-	In the Q-learning example in Section 4.1, is the effect of \epsilon considered? The discovery of non-myopic strategies might simply be based on chance. I would like the authors to include an investigation into the effect of this parameter.

Generally speaking, I appreciate the author’s thoughts on SIDS and how they approach revealing hidden incentives. Although I vote to reject this paper, I strongly encourage the authors to rewrite the paper, address all other issues that are noted during this peer review and resubmit.


**Experience Assessment:**

I have published in this field for several years.

**Review Assessment: Checking Correctness Of Derivations And Theory:**

I assessed the sensibility of the derivations and theory.

**Review Assessment: Checking Correctness Of Experiments:**

I assessed the sensibility of the experiments.

**Review Assessment: Thoroughness In Paper Reading:**

I read the paper thoroughly.

---

> ### Author Response · Authors · 2019-11-13
> **Authors' response: clarifying a few points and requesting elaboration**
>
> Thanks for you encouragement and detailed comments.
> We’d greatly appreciate further input on how to improve our submission!
>
> First, to clarify: context swapping is meant to remove *incentives* to induce (or prevent) distributional shift, but not to prevent or reduce SIDS, which may occur regardless of the learner’s incentives.  For example, Pennycook et al.[1] find evidence that the “illusory truth effect” can lead users to believe in “fake news”; this would happen regardless of whether an intelligent content recommendation system was trying to induce such an effect, or merely showed a user fake news articles because that’s what it predicted they would click on.
>
>
> Addressing your bullet points:
> - We really want our paper to be as clear as possible, and would love to know more specifically which sentences you found awkward or difficult to parse.
> - The distribution of “when the owner will wake up today” is shifted; the robot wakes them up in order to ensure that they will want coffee.  We can make this more explicit.  To be clear, our point is not that the robot will have difficulty learning in the presence of such a distributional shift; our point is that the robot having an incentive to produce such a shift is an alignment problem.
> - We agree this is overstated, and will soften the claim.  We’ve demonstrated this for PBT and REINFORCE (when considered as a meta-learning algorithm), only.  We believe it will hold true for a wide variety of meta-learning algorithms, but probably not all of them.
> - In fact, it is not the tuning of the learning rate that results in non-myopic behavior.  Rather, the EXPLOIT step of PBT is the main mechanism by which non-myopia is incentivized.  Appendix 3.1.2 walks through the mechanism.
> - Indeed, the choice of epsilon is important.  We use epsilon=0.1, and for much larger values of epsilon, non-myopic strategies are unstable and do not persist.  We will include more discussion and exploration of this choice.
> While there's certainly an element of chance (as our experiments demonstrate) as to whether the learner learns a stable non-myopic policy, we don’t think that invalidates the result; we think it is significant and surprising that Q-learning can yield a sub-optimal policy in 10/30 experiments.  Can you please explain why this is a concern for you?  Or elaborate on what you mean by “based on chance”?
>
>
> [1] Gordon Pennycook, Tyrone D Cannon, and David G. Rand. Prior exposure increases perceived accuracy of fake news. Journal of Experimental Psychology (forthcoming), 2019.

---

### Official Review · AnonReviewer3 · 2019-10-25
**Official Blind Review #3**

**Rating:** 1

**Review:**

The main idea of the paper: When using meta-learning there is an inherent incentive for the learner to win by making the task easier. The authors generalise this effect to a larger class of problems where the learning framework induces a set of Hidden Incentive for Distributional Shift (HIDS) and introduce Context Swapping, a HIDS mitigation technique. In the experimental section, the authors propos a HIDS unit test which then they employ to show that PBT (Population Based-Trainng), a popular meta-learning algorithm exhibits HIDS ant that context swapping helps fixing it.

Overall, I found the idea of the paper interesting, but the attempt to generalise the effect from meta-learning to general learning setups hard to follow and detracting from the overall value. I think the authors should have restricted their claims to PBT and PBT-like methods and follow-up with something more general in future work.

Furthermore, the notation and formaliation of the problem are incomplete:
    * the concept of ‘trajectory’ is introduced without being properly defined, though its crucial in the definition of the proposed HIDS mitigation approach
    * the context swapping algorithm description is not clearly motivated and explained, a diagram showing the learner shuffling would be quite helpful
    * in the HIDS unit-test section, the game theoretical setup is only partially explained, the defection and cooperation actions are not clearly linked to the HIDS
    * In Section 4.1.1 HIDS UNIT TEST EXPERIMENTAL RESULTS AND DISCUSSION Figure2 is refered for results without ever stating the task and the Figure itself does not mention it

       In terms of suggestions, I think the paper needs to go through a careful refactoring with an attention to technical details (careful concept definition, introduction of notation, clarity on experimental setup)

**Experience Assessment:**

I have read many papers in this area.

**Review Assessment: Checking Correctness Of Derivations And Theory:**

I assessed the sensibility of the derivations and theory.

**Review Assessment: Checking Correctness Of Experiments:**

I assessed the sensibility of the experiments.

**Review Assessment: Thoroughness In Paper Reading:**

I read the paper thoroughly.

---

> ### Author Response · Authors · 2019-11-13
> **Authors' response: requesting more feedback, and explaining why we included control experiments**
>
> We appreciate the feedback and would love to hear more detailed suggestions for improvements along the lines of your final paragraph.  Highlighting any parts of the experimental set-up that you found unclear, would be especially useful, since we believe we’ve already described it in sufficient detail.  We’ll be happy to make all of the specific changes you’ve suggested.
>
> Regarding “restrict[ing] the claims to PBT and PBT-like methods”, we’re sorry this wasn’t clear.  I assume you’re referring to the bottom half of page 6, where we discuss Q-learning and REINFORCE.  I’ll attempt to clarify this now, and request that you please let us know:
> 1) what if anything is still unclear to you,
> 2) whether you still think we shouldn’t have made such claims
> 3) what you think could be done to make this easier to follow.
> We included these additional experiments at the request of a previous reviewer, and we believe they shed some light on our results using PBT.  We chose these experiments precisely because we view these other algorithms as similar to PBT in important (but different) ways, and we believe these experiments serve as controls to support our hypotheses as to why PBT has this effect.
> We certainly believe that future work should aim for a more conclusive and general understanding of how choice of learning algorithms influence which incentives are pursued.  The connection we draw with meta-learning is just one example; including what we have already observed and hypothesized in these control experiments seems likely to help future researchers develop such understanding.

---

### Official Review · AnonReviewer1 · 2019-10-26
**Official Blind Review #1**

**Rating:** 1

**Review:**

The paper discusses concepts of self-induced distributional shift (SIDS) and the hidden incentives when using meta-learning algorithms. It then prescribes a unit-test to check whether there is hidden incentive for distributional shift (HIDS) in the algorithm and proposes to use context swapping to mitigate such phenomenon.

I am not very familiar with this series of research but I am wondering why the paper focuses on meta-learning and its connection to HIDS. Does it use meta-learning as a tool to identify HIDS? But from the description and experiments, it seems the paper is talking about how meta-learning itself leads to HIDS, for example, by comparing different hyper-parameter setting for meta-learning and PBT, it shows the unit-test is failing. So it seems like meta-learning itself leads to distributional shift?

Also I cannot fully appreciate the utility of this "unit-test". It is well known normally an interactive system that can change its inputs have distributional shift. What other information does this "unit-test" inform us? Similarly, how does "context swapping" mitigate distributional shift? From the experiments, it dumbs the meta-learning algorithms and make it pass the "unit-test", but I am not sure what other practical benefits it can bring to improve real systems.

Usually, a reinforcement learning algorithm can meaningfully mitigates the adverse effects of distributional shift by explicitly modeling this interactive process and evaluating rewards with considerations to distributional shift caused by different policies. It is difficult to see how the concepts discussed in the paper provide meaningful approaches to address the issue.

Overall, the paper touches the important question of distributional shift for machine learning systems but I find the concepts discussed in the paper, such as the focus on meta-learning, the "unit-test", and "context-swapping", less relevant to how we can really mitigate the issues in real systems or how it can provide additional insights about the problem.

**Experience Assessment:**

I do not know much about this area.

**Review Assessment: Checking Correctness Of Derivations And Theory:**

I assessed the sensibility of the derivations and theory.

**Review Assessment: Checking Correctness Of Experiments:**

I assessed the sensibility of the experiments.

**Review Assessment: Thoroughness In Paper Reading:**

I read the paper at least twice and used my best judgement in assessing the paper.

---

> ### Author Response · Authors · 2019-11-13
> **Authors' response: Answering questions and explaining relevance and relation to alternative approaches**
>
> Thank you for your thoughtful review.
> We’re glad that you find the topic important, and hope to clarify what we see as the relevance of our work.
> We’re striving to make our paper as clear as possible, and would greatly appreciate any further help in doing so, or identifying other ways to address the challenges of SIDS and especially HIDS.
>
> Addressing your questions:
>
>
> Q1: I am not very familiar with this series of research but I am wondering why the paper focuses on meta-learning and its connection to HIDS. Does it use meta-learning as a tool to identify HIDS?
> A1: We believe we are the first to study HIDS.   We think meta-learning provides a clear illustration of why HIDS might lead to easily-overlooked problems; meta-learning is often framed as a method of finding a better solution to a given problem, but in fact it can also change what counts as a good solution, and our impression is that many researchers find that surprising (sentence 1 of final paragraph of section 1).
>
> Q2: Does [we] use meta-learning as a tool to identify HIDS?
> A2: Yes, you can view it this way.  But the point is that one should be aware of which incentives are hidden/visible, and also be aware that seemingly innocuous changes in the learning algorithm can change that (sentence 3 of section 3.2).
>
> Q3: It is well known normally an interactive system that can change its inputs have distributional shift. What other information does this "unit-test" inform us?
> A3: We agree this is well known.   The unit test tells us whether a given learner is indifferent to such changes, or will actively seek to induce them.  Consider our example of content recommendation.  Content recommendation can change user interests whether or not the learner is seeking to induce such a shift, but we should be more concerned about algorithms that view changing user interests as a legitimate strategy to improve performance than those that are indifferent to such changes.
>
> Q4: Similarly, how does "context swapping" mitigate distributional shift? From the experiments, it dumbs the meta-learning algorithms and make it pass the "unit-test", but I am not sure what other practical benefits it can bring to improve real systems.
> A4: TODO: an example
> Context swapping doesn’t make the meta-learning algorithms less smart, it merely changes their incentives, aiming to remove incentives for distributional shift.  However, as we noticed in the content recommendation experiments, it doesn’t work well in situations where learners are unable to track any distributional shift which does occur.  In other words, it’s only a starting point for managing learners’ incentives, not a complete solution.
>
>
>
> Regarding your 2nd to last paragraph:
> First, I'm not precisely sure what you are suggesting as an alternative. Can you be more concrete, or provide an example?
> It seems like you are suggesting that RL algorithms can learn to model distributional shift in the environment and account for it.
> While this is true, this does not address the issue of whether/when an RL agent should view SIDS as a legitimate part of a solution strategy.  By default, RL algorithms aim to maximize returns by any means, viewing any form of SIDS as something which should be leveraged to drive up performance.
> We could try to address this by providing a reward function that penalizes only those SIDS we think are undesirable. For example, in the content recommendation setting, instead of seeking learners indifferent to changes in user preferences, we could attempt to provide the learner with a specification that would distinguish between good changes (e.g. based on informing users) and bad changes (e.g. based on manipulating or misinforming users).
> However, we have concerns about the scalability and tractability of this as a fully general approach, since it seems to rely on the learner having thorough knowledge of human preferences over different outcomes.  Such an approach may be impractical and error-prone, since it may require the reward to be a function of the entire history of interaction.  It may also be undesirably value-laden, since different users may have different ideas about what forms of influence are (il)legitimate.
> This is discussed briefly in paragraph 3 of the introduction.

---

> > ### Author Response · Authors · 2019-11-13
> > **Authors' response: Answering questions and explaining relevance and relation to alternative approaches (...continued)**
> >
> >
> > Regarding the final paragraph of your review on the relevance of our work:
> > There are already well-known issues related to SIDS, and works addressing them.  If I understand correctly, we are on the same page about SIDS, but you are skeptical that:
> > 1) HIDS presents new challenges that require new approaches.
> > and/or
> > 2) The challenges HIDS raised have (or will have) significant practical relevance.
> > and/or
> > 3) Our work makes useful progress in addressing challenges of HIDS.
> > It would help us to know which of the above describes your position!  Our primary aim is to explain HIDS, the kinds of problems we imagine it might lead to, and some ideas for how they could be addressed.  We hope that a clear exposition of the problems related to HIDS will motivate other researchers to come up with their own ideas for how to diagnose and address these problems in practice.
> >
> > While we agree that it’s unclear how significant our work is for current issues with real-world systems, we believe we provide important insights!  Two in particular are:
> > 1) Viewing the learning algorithm (and not just the loss/reward function) as an important aspect of specification.
> > 2) Learners trained with supervised learning + meta-learning can pursue incentives for SIDS (similarly to RL agents).
> > In general, we believe that understanding when and why incentives are hidden or revealed is an important and interesting scientific question that deserves attention, and hope to clearly communicate this question and our motivation for studying it.

---

> > > ### Comment · AnonReviewer1 · 2019-11-15
> > > **Reply**
> > >
> > > Thanks for your reply to my questions.
> > >
> > > My main points are that
> > >
> > > 1) In interactive systems, it is well known that there are some distributional shift with simple reasoning. We don't need a "unit-test" to verify that.
> > >
> > > 2) Practically, what matters is how we can reduce negative consequences from distributional shift. It is also helpful if we can leverage the distributional shift to achieve desirable outcomes.
> > >
> > > 3) To promote beneficial effects from distributional shift, we should focus on the design of reward function or metrics that align well with our desired goals. Just preventing an algorithm to induce distributional shift is less relevant.
> > >
> > > In particular, with context swapping, do we get better performing learning algorithms compared with not doing context swapping? If not, what is the benefit of context swapping? There are many dumb models, e.g. just predicting constants, that do not have incentives for distributional shifts, but they are not what we want.
> > >
> > > For example, in recommendation system, it is actually useful if we can account for and leverage the distributional shift and in the end achieve better rewards. One such case is the recent online deployment of reinforcement learning algorithms in Youtube [1].
> > >
> > > [1] Chen, M., Beutel, A., Covington, P., Jain, S., Belletti, F. and Chi, E.H., 2019, January. Top-k off-policy correction for a REINFORCE recommender system. In Proceedings of the Twelfth ACM International Conference on Web Search and Data Mining (pp. 456-464). ACM.

---

### Decision · Program_Chairs · 2019-12-19

**Decision:**

Reject

**Comment:**

The paper shows how meta-learning contains hidden incentives for distributional shift and how a technique called context swapping can help deal with this. Overall, distributional shift is an important problem, but the contributions made by this paper to deal with this, such as the introduction of unit-tests and context-swapping, is not sufficiently clear. Therefore, my recommendation is a reject.